# Ocular Tolerability of Bimatoprost 0.1 mg/mL Preservative-Free versus Bimatoprost 0.1 mg/mL with Benzalkonium Chloride or Bimatoprost 0.3 mg/mL Preservative-Free in Patients with Primary Open-Angle Glaucoma

**DOI:** 10.3390/jcm11123518

**Published:** 2022-06-19

**Authors:** Mariaelena Filippelli, Giuseppe Campagna, Nicola Ciampa, Gaetano Fioretto, Roberta Giannini, Pier Franco Marino, Roberto dell’Omo, Ciro Costagliola

**Affiliations:** 1Department of Medicine and Health Sciences, “V. Tiberio”, University of Molise, 86100 Campobasso, Italy; roberto.dellomo@unimol.it (R.d.); ciro.costagliola@unimol.it (C.C.); 2Department of Medical-Surgical Sciences and Translational Medicine, University of Rome “Sapienza”, 00185 Rome, Italy; gius.campagna@gmail.com; 3Department of Neurosciences, Reproductive Sciences and Dentistry, University of Naples Federico II, 80138 Naples, Italy; nicolaciampa3@gmail.com (N.C.); fioretto.gaetano@gmail.com (G.F.); alpapini@tiscali.it (P.F.M.); 4Department of Ophthalmology, San Camillo Hospital, 00152 Rome, Italy; robertagiannini1@gmail.com

**Keywords:** primary open-angle glaucoma, bimatoprost, benzalkonium chloride, break-up time (BUT) test, ocular surface disease index (OSDI), intraocular pressure (IOP)

## Abstract

This study aimed to evaluate whether the therapeutic switch from a formulation of Bimatoprost 0.1 mg/mL with benzalkonium chloride (BAK) or Bimatoprost 0.3 mg/mL preservative-free to a formulation of Bimatoprost 0.1 mg/mL preservative-free could improve eye surface conditions in patients with glaucoma; intraocular pressure (IOP) was also evaluated. All patients meeting the inclusion criteria were eligible for the therapeutic switch to Bimatoprost 0.1 mg/mL preservative-free. At each check visit, enrolled patients underwent a break-up time (BUT) test, an ocular surface disease index (OSDI) test, and a three-point tonometric curve. A total of 40 patients were enrolled (23 were in therapy with Bimatoprost 0.1 mg/mL with BAK and 17 with Bimatoprost 0.3 mg/mL preservative-free). Significant differences of OSDI and BUT between Bimatoprost 0.1 mg/mL with BAK at baseline vs. Bimatoprost 0.1 mg/mL preservative-free at 14 and 28 days (*p* < 0.0001 and *p* = 0.0003, respectively) were recorded. Similarly, significant differences of OSDI and BUT between Bimatoprost 0.3 mg/mL preservative-free at baseline vs. Bimatoprost 0.1 mg/mL preservative-free at 14 and 28 days (*p* < 0.0001 for both) were found. Bimatoprost 0.1 mg/mL preservative-free has a better tolerability profile associated with non-therapeutical inferiority in the control of IOP compared to the other Bimatoprost formulations.

## 1. Introduction

Glaucoma is defined as a group of irreversible, progressive optic neuropathies that can lead to severe visual field loss and blindness [1]. Primary open-angle glaucoma (POAG) is the most widespread type of glaucoma in European and African populations [2]. It is estimated that 57.5 million people worldwide suffer from POAG and it is expected that this number will reach 111.8 million by 2040 [3]. The reduction in light scattered by the retinal nerve fiber layer (RFNL) near the optic nerve head is assumed to be an early indicator of axonal degeneration and a sensitive way to identify glaucomatous damage [4,5]. Several risk factors for glaucoma onset are known, such as age, gender, family history of glaucoma, genetics, race (no white ethnicity), myopia, pseudoexfoliation, disc hemorrhage, vasospasm, systemic hypotension/hypertension, obstructive sleep apnea syndrome, smoking, and last but not least increased intraocular pressure (IOP) [6,7,8]. Among all these risk factors the main one has been shown to be elevated IOP. Moreover, IOP is also the only currently treatable risk factor [9]. Management of elevated IOP is usually started with medical therapy. The latter consists of β-blockers, carbonic anhydrase inhibitors, α-agonists, miotics, and prostaglandin analogs (PGs), which are the most potent ocular hypotensive medications used in the treatment of POAG [10]. The fifth edition of the European Glaucoma Society (EGS) reports that PGs are the most effective medication and they are usually recommended as first-choice treatment in POAG [11]. The PGs used for glaucoma therapy are latanoprost, bimatoprost, travoprost, and tafluprost. Several clinical trials and meta-analyses have compared the efficacy and tolerance of different PGs [10,12]. The meta-analysis of Tang et al. revealed that bimatoprost is more effective in controlling IOP compared to latanoprost following longer treatment (3 and 6 months), and is more effective compared to travoprost when used for three months in patients with POAG [10]. On the other hand, conjunctival hyperemia occurs more often in patients treated with bimatoprost and travoprost compared to those under latanoprost therapy. Moreover, a higher incidence of lashes growth has been reported in patients treated with bimatoprost [10,13]. In 2010, in order to improve bimatoprost tolerability, a new strength, bimatoprost 0.1 mg/mL eye drops, in solution, was released as an alternative to the bimatoprost 0.3 mg/mL eye drops in solution. Bimatoprost 0.1 mg/mL eye drops compared to bimatoprost 0.3 mg/mL have a higher concentration of benzalkonium chloride (BAK) to increase the ocular absorption of bimatoprost, thus allowing for a lower concentration of bimatoprost to be administered (0.1 mg/mL). This new formulation, with a reduced concentration of bimatoprost, achieves comparable IOP-lowering efficacy to the current authorized strength and an improved overall safety profile [14,15]. However, the toxic effects of BAK are well known, particularly its effects on the ocular surface [16]. Usual side effects are conjunctival hyperemia, decreased tear production, tear film instability, and superficial punctate keratitis. This can lead to ocular discomfort as a result of dry eye and inflammatory irritation [13,17,18]. To verify the occurrence of these detrimental effects of BAK, we conducted a study to observe whether the therapeutic switch from a formulation of bimatoprost 0.1 mg/mL with 200 ppm of BAK to a formulation of bimatoprost 0.1 mg/mL preservative-free can improve eye surface conditions. We also evaluated whether the therapeutic switch from a formulation of bimatoprost 0.3 mg/mL preservative-free to a formulation of bimatoprost 0.1 mg/mL preservative-free can still positively affect the status of the eye surface. Finally, the therapeutic effect on IOP of the different formulations examined was also evaluated.

## 2. Materials and Methods

This open observational study was conducted in adherence to the tenets of the Declaration of Helsinki and obtained ethical approval from the Scientific Technical Committee (CTS) of the Department of Medicine and Health Sciences “V. Tiberio” of Molise University, Campobasso, Italy. Written informed consent was obtained from all participants after a detailed description of the procedure used and of the work’s aim. The study was conducted at the Department of Medicine and Health Sciences “V. Tiberio” of Molise University, Campobasso (Italy), at the Department of Neurosciences, Reproductive Sciences, and Dentistry, University of Naples Federico II, Naples (Italy), and at the Department of Ophthalmology, San Camillo Hospital, Rome (Italy) from July 2021 to October 2021. Patients of both sexes, aged 45–70, affected by POAG afferent to the glaucoma service were selected for the study. The patients who met the inclusion criteria, after the initial examination and the survey of the parameters established by the protocol, were instructed on the experimental procedure and on the planned controls. During the experimental period, no intake of products with antibiotics, anti-inflammatory, or other eye drops (tear substitutes, etc.) was allowed. The possible consumption of these compounds was reported in the “Data Collection Sheet” and justified the exclusion from the trial. The study was subdivided into two sub-studies. In the first (Study A) were enlisted 23 patients in monotherapy with Bimatoprost 0.1 mg/mL with 200 ppm of BAK (Lumigan^®^ 0.1 mg/mL, Allergan, Inc., Dublin, Ireland). In the second (Study B) there were 17 patients in monotherapy with Bimatoprost 0.3 mg/mL preservative-free (Lumigan^®^ 0.3 mg/mL, Allergan, Inc.). The inclusion and exclusion criteria were the same for both studies.

Inclusion criteria:

Monotherapy with Bimatoprost 0.1 mg/mL with 200 ppm of BAK (study A). Monotherapy with Bimatoprost 0.3 mg/mL preservative-free (study B).

Ocular test surface disease index (OSDI) > 22, break-up time test (BUT) < 10 s, IOP between 14 and 21 mmHg in therapy, stable perimetric indices, no previous cataract surgery, no previous diagnosis of dry eye disease, no diseases of the thyroid, no systemic therapies capable of altering normal lacrimal production (e.g., β-blockers) and pachimetry between 490 and 560 μm.

Exclusion criteria:

Patients with ocular diseases or requiring treatments that may impair the assessment of the treatments; in particular, patients who should be given preparations for antibiotic and/or anti-inflammatory activities, patients with systemic diseases (diabetes, thyroid disease, hypertension, hemopathies, etc.), unreliable patients regarding compliance and compliance with scheduled checks, addition of an artificial tear to hypotonizing therapy in place, onset of seasonal allergic symptoms, patients who during the observation period develop ocular or systemic pathological events affecting the continuity of the study, or undergo operations compromising the efficacy of the treatment and need to add hypotonizing eye drops to manage IOP.

All patients meeting the inclusion criteria were considered eligible for the therapeutic switch to Bimatoprost 0.1 mg/mL preservative-free (Bimanext^®^, FB VISION S.*p*.A) once a day. At baseline, enrolled patients were in therapy with Bimatoprost 0.1 mg/mL with 200 ppm of BAK (study A) or Bimatoprost 0.3 mg/mL preservative-free (study B). For included patients, four checks were scheduled. The clinical evaluations were performed at T0 (basal visit); T1 (beginning treatment after 7 days of washout [11] from the previous drug); T2 (14 days from the start of treatment); T3 (28 days from the start of treatment). At each check, the following examinations were carried out: OSDI; BUT; three-point tonometric curve (8.00–13.00–18.00) by non-contact tonometer (Nidek Tonoref III). The status of the ocular surface was evaluated both subjectively throughout the perception of ocular discomfort measured with the OSDI test, and objectively through the BUT test. The BUT test was assessed according to the guidelines published in the report of the DEWS 2007 using Minims fluorescein sodium 2.0% eye drops [19]. Measurements were repeated three times, using the mean value. In addition, the overlap of the two test preparations in the control of IOP was verified by the three-point tonometric curve (Figure 1). The primary endpoint of both studies (A and B) was the improvement of at least eight points (study A) or six points (study B) in the OSDI test, whereas the secondary endpoint was the evaluation of IOP variations. Any side effect reported by the patient, even if not attributable to the treatment conducted, was transcribed on the “Data Collection Sheet”.

### 2.1. Sample Size Study A

The sample size was determined by assuming a clinically reliable difference in OSDI, d = 8, among the same patients who were previously treated with Bimatoprost 0.1 mg/mL with BAK and then with Bimatoprost 0.1 mg/mL preservative-free patients with a standard deviation σ = 11.10 [18], which gives a sample size *n* =19 with α = 0.05 and power = 80%; finally considering a drop-out of 20%, *n* = 23 patients were recruited for the final study. The sample size was determined using the *proc power pairedmeans test = diff* procedure performed with SAS v.9.4 (SAS Institute Inc., Cary, NC, USA).

### 2.2. Sample Size Study B

The sample size was determined by assuming a clinically reliable difference in OSDI, d = 6, among the same patients who will first be treated with Bimatoprost 0.3 mg/mL preservative-free and then with Bimatoprost 0.1 mg/mL preservative-free patients with a standard deviation σ = 6.25 [20], which gives a sample size *n* = 14 with α = 0.05 and power = 80%; finally considering a drop-out of 20%, *n* = 17 patients were recruited for the final study. The sample size was determined using the *proc power pairedmeans test* = *diff* procedure performed with SAS v.9.4 (SAS Institute Inc., Cary, NC, USA).

### 2.3. Statistical Analysis

Continuous variables were presented as mean ± standard deviation (SD) and 95% confidence interval (CI). Categorical variables were expressed as absolute frequencies and percentages—*n* (%).

A generalized linear mixed model (GLIMMIX) for repeated measures with normal distribution was used to verify the differences of OSDI, BUT, and IOP between Bimatoprost 0.1 mg/mL with BAK at baseline/Bimatoprost 0.3 mg/mL preservative-free at baseline, and Bimatoprost 0.1 mg/mL preservative-free at 14 and 28 days. Post-hoc analysis was performed using the Tukey method. This method was used to correct the *p* values in the presence of multiple comparisons. Normality residuals were tested with the Shapiro–Wilk test and checking the Q-Q (quantile–quantile) plot. Homoscedasticity was evaluated by checking the studentized residuals vs. fitted values plot. A value of *p* < 0.05 was considered statistically detectable. Continuous variables were represented by violin and box-plot graphs. A violin plot includes all the data that are in a box-plot but it is more informative. A box-plot shows the mean/median and interquartile ranges, whereas the violin plot shows the full distribution of the data. A violin plot shows possible peaks, their position, and relative amplitude. All statistical analyses were performed using SAS v.9.4 and JMP PRO 16.1 (SAS Institute Inc., Cary, NC, USA). For statistical analysis the right eye of each enrolled patient was always taken into consideration.

## 3. Results

### 3.1. Study A

A total of 23 patients, 12 male (52.17%) and 11 (47.83%) female, with a mean age 60.87 ± 11.45 (95% CI: 55.92 to 65.82) were included and all completed the study according to the protocol. Table 1 shows significant differences of OSDI and BUT between Bimatoprost 0.1 mg/mL with BAK at baseline vs. Bimatoprost 0.1 mg/mL preservative-free at 14 and 28 days (*p* < 0.0001 and *p* = 0.0003, respectively). OSDI and BUT differences between Bimatoprost 0.1 mg/mL with BAK at baseline vs. Bimatoprost 0.1 mg/mL preservative-free at 14 and 28 days (*p* < 0.0001 for both) and (*p* = 0.002 and *p* = 0.0006 respectively) were highlighted by post-hoc analysis (Figure 2 and Figure 3). IOP was distributed equally between treatments (Table 1 and Figure 4). Table 2 shows the absence of differences between the treatments relatively to three times of measurement (8.00–13.00–18.00) of IOP. Lastly, adverse events recorded during the study are reported in Table 3.

### 3.2. Study B

A total of 17 patients, 8 (47.06%) male, and 9 (52.94%) female, with a mean age of 62.71 ± 6.50 (95% CI: 59.36 to 66.05) were included and all finished the study according to the protocol. Table 4 shows significant differences of OSDI and BUT between Bimatoprost 0.3 mg/mL preservative-free at baseline vs. Bimatoprost 0.1 mg/mL preservative-free at 14 and 28 days (*p* < 0.0001 for both). OSDI and BUT differences between Bimatoprost 0.3 mg/mL with preservative-free at baseline vs. Bimatoprost 0.1 mg/mL preservative-free at 14 and 28 days (*p* < 0.0001 for all) and Bimatoprost 0.1 mg/mL preservative-free at 14 vs. Bimatoprost 0.1 mg/mL preservative-free at 28 days, *p* = 0.01 and *p* < 0.0001, respectively, were highlighted by post-hoc analysis (Figure 5 and Figure 6). IOP is distributed equally between treatments (Table 4 and Figure 7). Table 5 shows the absence of differences between the treatments relatively to three times of measurement (8, 13, and 18 h) of IOP. Lastly, the adverse events recorded during the study are reported in Table 6.

## 4. Discussion

The long-term use of anti-glaucoma drugs has been associated with toxic and inflammatory changes in the ocular surface, which may be due either to the preservative, in particular benzalkonium chloride, or directly to the hypotonizing drug, especially prostaglandins and prostamides [18,21]. Furthermore, the turn-over of the preservative is very slow, and the quaternary ammonium molecules can remain in the ocular tissues even up to seven days [22,23]. Three main mechanisms of secondary toxicity to BAK have been described: (i) loss of stability of the precorneal tear film due to detergent action; (ii) direct toxic effects at the level of corneal and conjunctival epithelia, and immune-allergic effects; (iii) decrease of the stability of the precorneal tear film, with increased evaporation both directly (due to their surfactant properties and the cleaning effects of the lipid layer), and indirectly (through the decrease of the mucous cells of the conjunctival epithelium) [23,24,25,26]. In our study, a significant difference of OSDI and BUT was found in both studies. The OSDI questionnaire is designed to subjectively measure the frequency of specific symptoms and their impact on vision-related tasks of daily life [27]. BUT objectively verifies the time to tear breakup while the patient eye is open under the cobalt blue light of a slit-lamp. It is evident that the OSDI results are strengthened by the enhancement of BUT. Hence, the use of eyedrops not containing BAK allows the improvement of the ocular surface. Some studies have shown that after repeated instillations BAK reaches the trabecular meshwork and promotes its degeneration, as reported in glaucomatous patients (trabecular apoptosis, oxidative stress, induction of inflammatory chemokines). These findings corroborate the hypothesis that antiglaucoma eye drops, through the toxicity of their preservative, may induce further long-term trabecular degeneration and therefore increase outflow resistance, reducing the effectiveness of IOP-lowering agents [16,28]. These alterations, along with causing obvious discomfort to the patient, can also seriously affect the outcomes of surgery of trabeculectomy or trabeculoplasty. Therefore, in the case of chronic therapies with eye drops containing preservatives, it may be useful to assess the chance of a therapeutic shift towards BAK-free solutions [29]. In addition, there are also effects directly related to certain classes of medication. Prostaglandins and/or prostamides are responsible for conjunctival hyperemia, hypertrichosis, change of iris color (pigmentation), hyperpigmentation of the periocular area, burning, and sensation of foreign body [30]. When chronically applied, it may be useful to evaluate a therapeutic change towards solutions with a lower concentration of the active drug, which, however, exhibits a non-inferiority in the IOP control. This goal had already been achieved with the use of Bimatoprost 0.1 mg/mL instead of Bimatoprost 0.3 mg/mL, as confirmed also by our findings. Figus et al. showed that Bimatoprost 0.1 mg/mL eye drops improve ocular discomfort with respect to Bimatoprost 0.3 mg/mL eye drops, demonstrating a significant amelioration of all functional parameters [15]. Although it is easy to understand that a reduction in prostaglandin concentration favors a greater tolerability profile, it is not so easy to understand the therapeutic non-inferiority to the reduction of the drug concentration. This aspect had already been analyzed when Bimatoprost 0.1 mg/mL with BAK was introduced into the market. Surprisingly, the reduction of the active principle does not correspond to reduced therapeutic efficacy. Myers et al. showed that of the available Bimatoprost formulations, Bimatoprost 0.1 mg/mL had the more favorable efficacy and safety profile [31]. Similarly, in both our studies (Study A and Study B) no statistically significant difference between the two formulations in the control of the IOP was recorded (Study A: *p* = 0.92; Study B: *p* = 0.97). Therefore, also in our study Bimatoprost 0.1 mg/mL was not inferior to Bimatoprost 0.3 mg/mL in terms of therapeutic efficacy. However, to the best of our knowledge, it is the first time that Bimatoprost 0.1 mg/mL preservative-free has been tested. Our study demonstrates that the therapeutic efficacy of Bimatoprost 0.1 mg/mL preservative-free is not inferior neither to Bimatoprost 0.3 mg/mL preservative -free nor to Bimatoprost 0.1 mg/mL with BAK. Finally, considering that the primary endpoint of the study was the improvement by at least eight points (study A) or six points (study B) in the OSDI test, whereas the secondary endpoint was the evaluation of IOP variations, it is possible to state that both endpoints have been achieved.

## 5. Conclusions

The use of Bimatoprost 0.1 mg/mL preservative-free had a better tolerability profile associated with non-therapeutical inferiority in the control of IOP, compared to the other formulations examined. In addition, it is essential to highlight the crucial clinical impact of the results obtained. In fact, the improvement of the OSDI, corresponding to an improvement in the comfort of the patient, also leads to greater and better therapeutic compliance. Therefore, in the therapeutic scenario of glaucoma, the presence of a prostaglandin is not only effective but also well-tolerated by patients; with the reduction of harmful effects related to chronic use of BAK, it is helpful for patients and ophthalmologists.

## Figures and Tables

**Figure 1 jcm-11-03518-f001:**
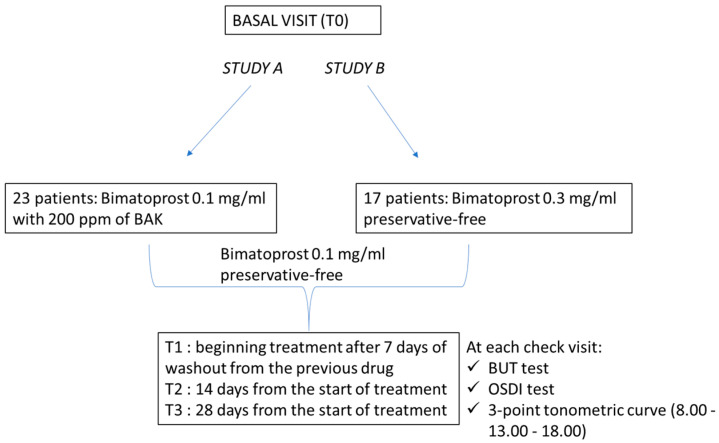
Flow chart of the study.

**Figure 2 jcm-11-03518-f002:**
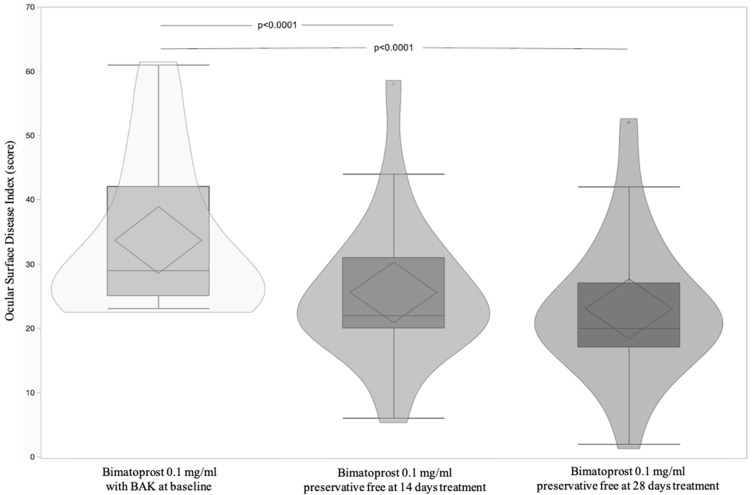
Violin and box-plots of the ocular surface index by Bimatoprost 0.1 mg/mL with BAK at baseline and the Bimatoprost 0.1 mg/mL preservative-free at 14 and 28 days.

**Figure 3 jcm-11-03518-f003:**
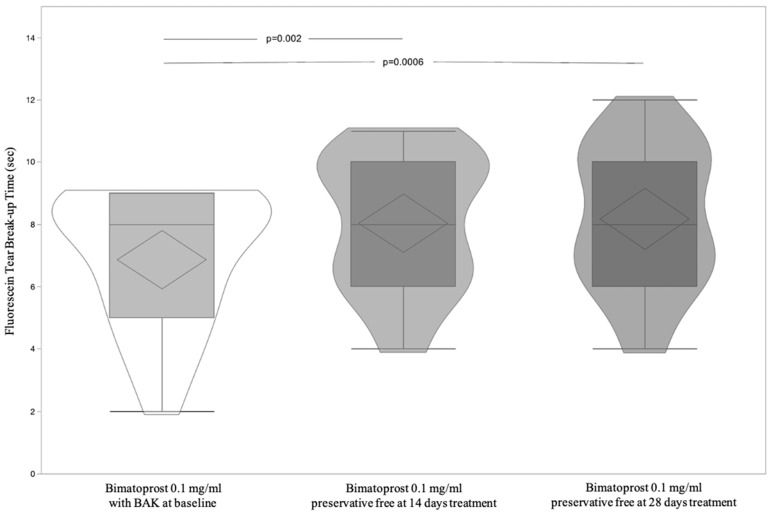
Violin and box-plots of fluorescein tear break-up time by Bimatoprost 0.1 mg/mL with BAK at baseline and Bimatoprost 0.1 mg/mL preservative-free at 14 and 28 days.

**Figure 4 jcm-11-03518-f004:**
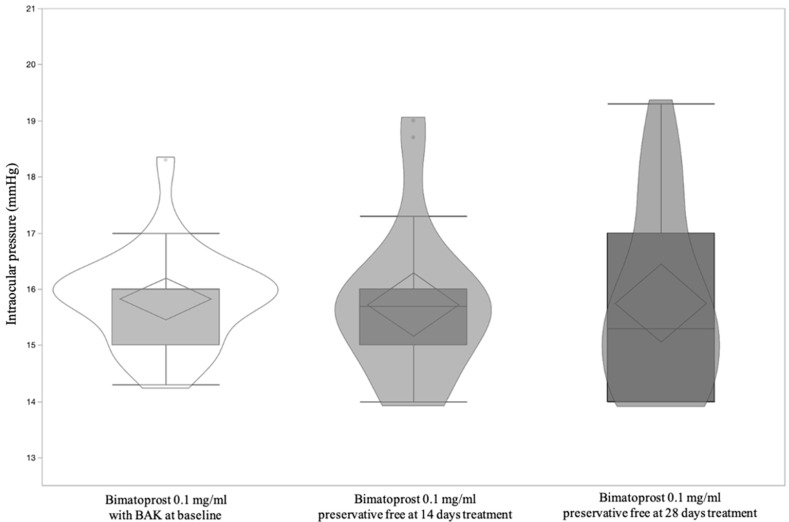
Violin and box-plots of intraocular pressure by Bimatoprost 0.1 mg/mL with BAK at baseline and Bimatoprost 0.1 mg/mL preservative-free at 14 and 28 days.

**Figure 5 jcm-11-03518-f005:**
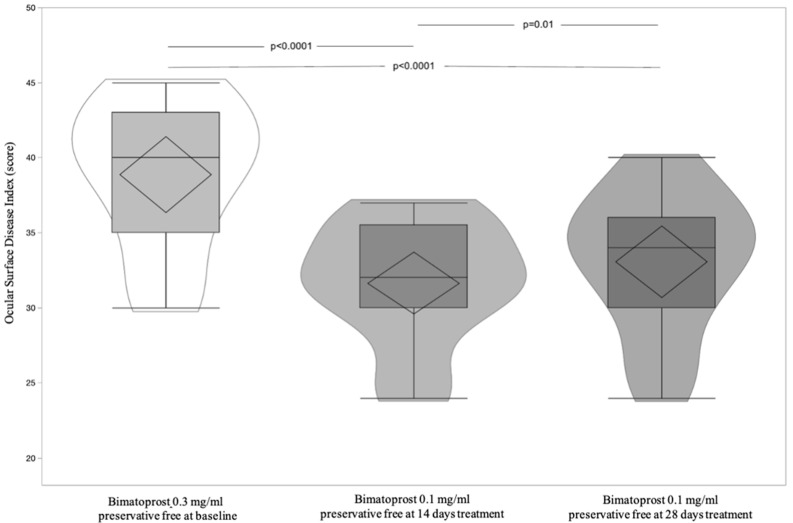
Violin and box-plots of the ocular surface index by Bimatoprost 0.3 mg/mL preservative-free at baseline and Bimatoprost 0.1 mg/mL preservative-free at 14 and 28 days.

**Figure 6 jcm-11-03518-f006:**
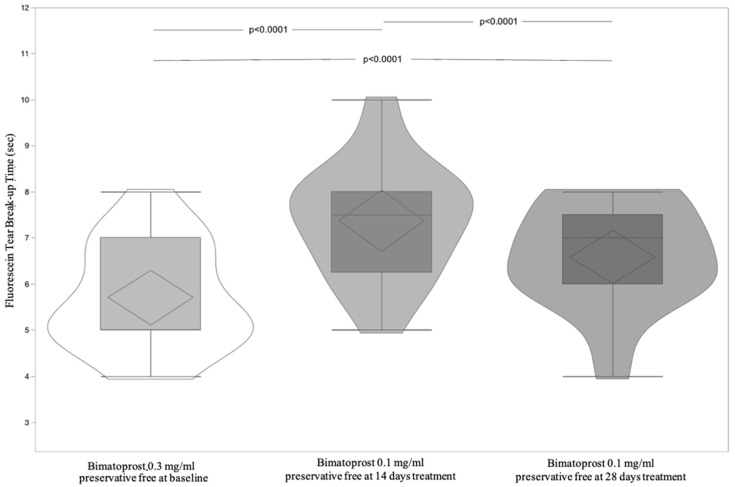
Violin and box-plots of fluorescein tear break-up time by Bimatoprost 0.3 mg/mL preservative-free at baseline and Bimatoprost 0.1 mg/mL preservative-free at 14 and 28 days.

**Figure 7 jcm-11-03518-f007:**
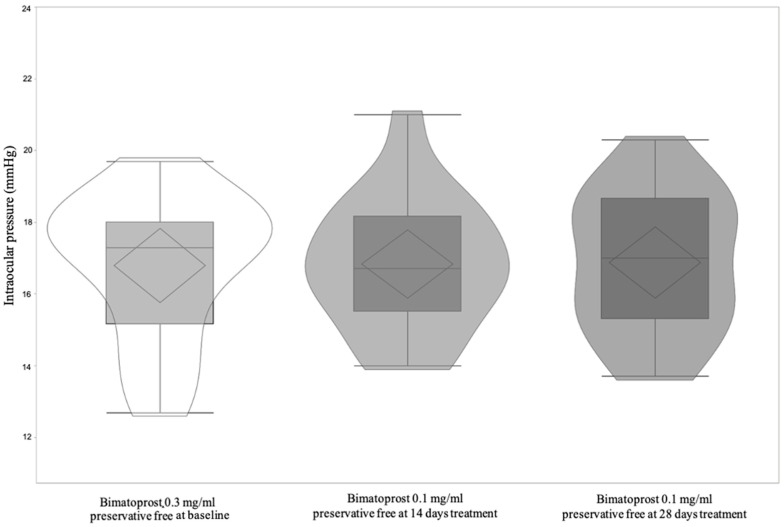
Violin and box-plots of intraocular pressure by Bimatoprost 0.3 mg/mL preservative-free at baseline and Bimatoprost 0.1 mg/mL preservative-free at 14 and 28 days.

**Table 1 jcm-11-03518-t001:** Comparisons between Bimatoprost 0.1 mg/mL with BAK at baseline and Bimatoprost 0.1 mg/mL preservative-free at 14 and 28 days relative to ocular parameters.

Parameter	Bimatoprost 0.1 mg/mL with BAK at Baseline Mean ± SD (95% CI)	Bimatoprost 0.1 mg/mL Preservative-Free at 14 Days Mean ± SD (95% CI)	Bimatoprost 0.1 mg/mL Preservative-Free at 28 Days Mean ± SD (95% CI)	*p*
OSDI (score)	33.74 ± 12.02 (28.54 to 38.93)	25.61 ± 10.88 (20.9 to 30.31)	23.00 ± 10.70 (18.37 to 27.63)	**<0.0001**
BUT (sec)	6.87 ± 2.16 (5.94 to 7.80)	8.04 ± 2.16 (7.11 to 8.98)	8.17 ± 2.29 (7.18 to 9.16)	**0.0003**
IOP (mmHg)	15.83 ± 0.86 (15.45 to 16.20)	15.73 ± 1.29 (15.17 to 16.29)	15.75 ± 1.61 (15.05 to 16.45)	0.92

Abbreviations: OSDI—ocular surface disease index; BUT—break-UP TIME; IOP—intraocular pressure; Post-hoc analysis: OSDI: Bimatoprost 0.1 mg/mL with BAK at baseline vs. Bimatoprost 0.1 mg/mL preservative-free at 14 days, *p* < 0.0001; Bimatoprost 0.1 mg/mL with BAK at baseline vs. Bimatoprost 0.1 mg/mL preservative-free at 28 days, *p* < 0.0001; Bimatoprost 0.1 mg/mL preservative-free at 14 days vs. Bimatoprost 0.1 mg/mL preservative-free at 28 days, *p* = 0.07; FT-BUT: Bimatoprost 0.1 mg/mL with BAK at baseline vs. Bimatoprost 0.1 mg/mL preservative-free at 14 days, *p* = 0.002; Bimatoprost 0.1 mg/mL with BAK at baseline vs. Bimatoprost 0.1 mg/mL preservative-free at 28 days, *p* = 0.0006; Bimatoprost 0.1 mg/mL preservative-free at 14 days vs. Bimatoprost 0.1 mg/mL preservative-free at 28 days, *p* = 0.91.

**Table 2 jcm-11-03518-t002:** Comparisons between the treatments and the Intraocular pressure measurements at 8, 13, and 18 h.

Parameter	Bimatoprost 0.1 mg/mL with BAK at Baseline Mean ± SD (95% CI)	Bimatoprost 0.1 mg/mL Preservative-Free at 14 Days Mean ± SD (95% CI)	Bimatoprost 0.1 mg/mL Preservative-Free at 28 Days Mean ± SD (95% CI)
IOP, mmHg			
At 8 h	16.00 ± 1.04 (15.55 to 16.45)	15.83 ± 1.40 (15.22 to 16.43)	15.65 ± 1.75 (14.90 to 16.41)
At 13 h	15.74 ± 0.86 (15.38 to 16.11)	15.70 ± 1.26 (15.15 to 16.24)	15.70 ± 1.58 (15.01 to 16.38)
At 18 h	15.75 ± 1.25 (15.20 to 16.28)	15.65 ± 1.50 (15.01 to 16.30)	15.91 ± 1.70 (15.18 to 16.65)

Abbreviations: IOP—intraocular pressure; Post-hoc analysis: At 8 h: Bimatoprost 0.1 mg/mL with BAK at baseline vs. Bimatoprost 0.1 mg/mL preservative-free at 14 days, *p* = 0.14; Bimatoprost 0.1 mg/mL with BAK at baseline vs. Bimatoprost 0.1 mg/mL preservative-free at 28 days, *p* = 0.22; Bimatoprost 0.1 mg/mL preservative-free at 14 days vs. Bimatoprost 0.1 mg/mL preservative-free at 28 days, *p* = 0.54. At 13 h: Bimatoprost 0.1 mg/mL with BAK at baseline vs. Bimatoprost 0.1 mg/mL preservative-free at 14 days, *p* = 0.88; Bimatoprost 0.1 mg/mL with BAK at baseline vs. Bimatoprost 0.1 mg/mL preservative-free at 28 days, *p* = 0.88; Bimatoprost 0.1 mg/mL preservative-free at 14 days vs. Bimatoprost 0.1 mg/mL preservative-free at 28 days, *p* = 1.00. At 18 h: Bimatoprost 0.1 mg/mL with BAK at baseline vs. Bimatoprost 0.1 mg/mL preservative-free at 14 days, *p* = 0.76; Bimatoprost 0.1 mg/mL with BAK at baseline vs. Bimatoprost 0.1 mg/mL preservative-free at 28 days, *p* = 0.53; Bimatoprost 0.1 mg/mL preservative-free at 14 days vs. Bimatoprost 0.1 mg/mL preservative-free at 28 days, *p* = 0.35

**Table 3 jcm-11-03518-t003:** Adverse event totals recorded between Bimatoprost 0.1 mg/mL with BAK at baseline and Bimatoprost 0.1 mg/mL preservative-free at 14 and 28 days.

Adverse Event	Bimatoprost 0.1 mg/mL with BAK at Baseline *n* (%)	Bimatoprost 0.1 mg/mL Preservative-Free at 14 Days *n* (%)	Bimatoprost 0.1 mg/mL Preservative-Free at 28 Days *n* (%)
**Hyperemia**			
absent	5 (21.74)	7 (30.43)	11 (47.83)
very mild	7 (30.43)	10 (43.48)	9 (39.13)
mild	9 (39.13)	5 (21.74)	3 (13.04)
severe/serious	2 (8.70)	1 (4.35)	0 (0.00)
**Photophobia**			
absent	7 (30.43)	14 (60.87)	16 (69.57)
very mild	11 (47.83)	8 (34.78)	6 (26.09)
mild	4 (17.39)	1 (4.35)	1 (4.35)
severe/serious	1 (4.35)	0 (0.00)	0 (0.00)
**Tearing**			
absent	4 (17.39)	7 (30.43)	11 (47.83)
very mild	13 (56.52)	16 (69.57)	12 (52.17)
mild	6 (26.09)	0 (0.00)	0 (0.00)
severe/serious	0 (0.00)	0 (0.00)	0 (0.00)
**Pain**			
0	17 (73.91)	21 (91.30)	22 95.65)
1	3 (13.04)	0 (0.00)	1 (4.35)
2	2 (8.70)	2 (8.70)	0 (0.00)
3	0 (0.00)	0 (0.00)	0 (0.00)
4	1 (4.35)	0 (0.00)	0 (0.00)

**Table 4 jcm-11-03518-t004:** Comparisons between Bimatoprost 0.3 mg/mL preservative-free at baseline and Bimatoprost 0.1 mg/mL preservative-free at 14 and 28 days relative to ocular parameters.

Parameter	Bimatoprost 0.3 mg/mL Preservative-Free at Baseline Mean ± SD (95% CI)	Bimatoprost 0.1 mg/mL Preservative-Free at 14 Days Mean ± SD (95% CI)	Bimatoprost 0.1 mg/mL Preservative-Free at 28 Days Mean ± SD (95% CI)	*p*
OSDI (score)	38.88 ± 4.95 (36.34 to 41.43)	31.65 ± 4.03 (29.57 to 33.72)	33.06 ± 4.63 (30.68 to 35.44)	**<0.0001**
BUT (sec)	5.71 ± 1.16 (5.11 to 6.30)	7.53 ± 1.37 (6.82 to 8.24)	6.59 ± 1.12 (6.01 to 7.16)	**<0.0001**
IOP (mmHg)	16.78 ± 2.02 (15.75 to 17.83)	16.84 ± 1.86 (15.89 to 17.80)	16.87 ± 1.95 (15.87 to 17.87)	0.97

Abbreviations: OSDI—ocular surface disease index; BUT—break-up time; IOP—intraocular pressure; Post-hoc analysis: OSDI: Bimatoprost 0.3 mg/mL preservative-free at baseline vs. Bimatoprost 0.1 mg/mL preservative-free at 14 days, *p* < 0.0001; Bimatoprost 0.3 mg/mL preservative-free at baseline vs. Bimatoprost 0.1 mg/mL preservative-free at 28 days, *p* < 0.0001; Bimatoprost 0.1 mg/mL preservative-free at 14 days vs. Bimatoprost 0.1 mg/mL preservative-free at 28 days, *p* = 0.01; FT-BUT: Bimatoprost 0.3 m preservative-free at baseline vs. Bimatoprost 0.1 mg/mL preservative-free at 14 days, *p* < 0.0001; Bimatoprost 0.3 mg/mL preservative-free at baseline vs. Bimatoprost 0.1 mg/mL preservative-free at 28 days, *p* < 0.0001; Bimatoprost 0.1 mg/mL preservative-free at 14 days *vs.*. Bimatoprost 0.1 mg/mL preservative-free at 28 days, *p* < 0.0001.

**Table 5 jcm-11-03518-t005:** Comparisons between the treatments and the intraocular pressure measurements at 8, 13, and 18 h.

Parameter	Bimatoprost 0.3 mg/mL Preservative-Free at Baseline Mean ± SD (95% CI)	Bimatoprost 0.1 mg/mL Preservative-Free at 14 Days Mean ± SD (95% CI)	Bimatoprost 0.1 mg/mL Preservative-Free at 28 Days Mean ± SD (95% CI)
IOP, mmHg			
At 8 h	17.53 ± 2.40 (16.29 to 18.76)	17.11 ± 1.93 (16.12 to 18.11)	17.41 ± 2.43 (16.16 to 18.66)
At 13 h	16.12 ± 2.06 (15.06 to 17.18)	16.65 ± 1.80 (15.72 to 17.57)	16.47 ± 1.62 (15.63 to 17.30)
At 18 h	16.71 ± 2.23 (15.56 to 17.85)	16.76 ± 2.28 (15.59 to 17.94)	16.76 ± 2.19 (15.64 to 17.89)

Abbreviations: IOP—intraocular pressure; Post-hoc analysis: At 8 h: Bimatoprost 0.3 mg/mL preservative-free at baseline vs. Bimatoprost 0.1 mg/mL preservative-free at 14 days, *p* = 0.35; Bimatoprost 0.3 mg/mL preservative-free at baseline vs. Bimatoprost 0.1 mg/mL preservative-free at 28 days, *p* = 0.79; Bimatoprost 0.1 mg/mL preservative-free at 14 days vs. Bimatoprost 0.1 mg/mL preservative-free at 28 days, *p* = 0.51. At 13 h: Bimatoprost 0.3 mg/mL preservative-free at baseline vs. Bimatoprost 0.1 mg/mL preservative-free at 14 days, *p* = 0.23; Bimatoprost 0.3 mg/mL preservative-free at baseline vs. Bimatoprost 0.1 mg/mL preservative-free at 28 days, *p* = 0.35; Bimatoprost 0.1 mg/mL preservative-free at 14 days vs. Bimatoprost 0.1 mg/mL preservative-free at 28 days, *p* = 0.79. At 18 h: Bimatoprost 0.3 mg/mL preservative-free at baseline vs. Bimatoprost 0.1 mg/mL preservative-free at 14 days, *p* = 0.59; Bimatoprost 0.3 mg/mL preservative-free at baseline vs. Bimatoprost 0.1 mg/mL preservative-free at 28 days, *p* = 0.89; Bimatoprost 0.1 mg/mL preservative-free at 14 days vs. Bimatoprost 0.1 mg/mL preservative-free at 28 days, *p* = 0.69.

**Table 6 jcm-11-03518-t006:** Adverse event totals recorded between Bimatoprost 0.3 mg/mL preservative-free at baseline and Bimatoprost 0.1 mg/mL preservative-free at 14 and 28 days.

Adverse Event	Bimatoprost 0.3 mg/mL Preservative-Free at Baseline *n* (%)	Bimatoprost 0.1 mg/mL Preservative-Free at 14 Days *n* (%)	Bimatoprost 0.1 mg/mL Preservative-Free at 28 Days *n* (%)
**Hyperemia**			
absent	0 (0.00)	0 (0.00)	1 (5.88)
very mild	4 (23.53)	12 (70.59)	9 (52.94)
mild	9 (52.94)	5 (19.41)	6 (35.29)
severe/serious	4 (23.53)	0 (0.00)	1 (5.88)
**Photophobia**			
absent	6 (35.29)	8 (47.06)	5 (29.41)
very mild	8 (47.06)	8 (47.06)	11 (64.71)
mild	3 (17.65)	1 (5.88)	1 (5.88)
severe/serious	0 (0.00)	0 (0.00)	0 (0.00)
**Tearing**			
absent	2 (11.76)	2 (11.76)	3 (17.65)
very mild	3 (17.65)	12 (70.59)	10 (58.62)
mild	12 (70.59)	3 (17.65)	3 (17.65)
severe/serious	0 (0.00)	0 (0.00)	1 (5.88)
**Pain**			
0	10 (58.62)	17 (100.00)	16 (94.12)
1	6 (35.29)	0 (0.00)	1 (5.88)
2	1 (5.88)	0 (0.00)	0 (0.00)
3	0 (0.00)	0 (0.00)	0 (0.00)

## Data Availability

Not applicable. The original contributions presented in the study are included in the article, further inquiries can be directed to the corresponding author/s.

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
