# Peer review of "Ocular Tolerability of Bimatoprost 0.1 mg/mL Preservative-Free versus Bimatoprost 0.1 mg/mL with Benzalkonium Chloride or Bimatoprost 0.3 mg/mL Preservative-Free in Patients with Primary Open-Angle Glaucoma"

_jcm, 2022, doi:10.3390/jcm11123518_

Round 1
Reviewer 1 Report
Filippelli et al. investigated the efficacy of Bimatoprost 0.1 mg/ml preservative free compared with other Bimatoprost formulations. There are several suggestions to be considered for the further process.
#1. Please edit minor errors for English. For instance, some sentences start with a lowercase letter. Also, some abbreviations are repeatedly full-spelled out.
#2. The inclusion criteria seem strange. For instance, the authors included those with no history of DED but those with BUT <10 sec. Please check carefully with the baseline inclusion criteria, especially for the ocular surface conditions.
#3. Although the author presented the tolerability profile associated with non-therapeutical inferiority, the baseline IOP was between 14-21 mmHg in the inclusion criteria. Therefore, the study subjects had mild glaucoma (well-controlled patients), which might not be the targeted population for using Bimatoprost. As a consequence, we cannot infer the inferiority of IOP decrease by differential concentration of Bimatoprost.
Author Response
#1. Please edit minor errors for English. For instance, some sentences start with a lowercase letter. Also, some abbreviations are repeatedly full-spelled out.
Thank you for the note. We responded to your request.
#2. The inclusion criteria seem strange. For instance, the authors included those with no history of DED but those with BUT <10 sec. Please check carefully with the baseline inclusion criteria, especially for the ocular surface conditions.
Thank you for your observation. Indeed, although BUT is not high, none of the patients included had a history or complained of DED symptoms.
#3. Although the author presented the tolerability profile associated with non-therapeutical inferiority, the baseline IOP was between 14-21 mmHg in the inclusion criteria. Therefore, the study subjects had mild glaucoma (well-controlled patients), which might not be the targeted population for using Bimatoprost. As a consequence, we cannot infer the inferiority of IOP decrease by differential concentration of Bimatoprost.
Thanks for the note. We selected all patients who, in accordance with the guidelines mentioned in the paper, were in therapy with Bimatoprost as the first therapeutic approach. Moreover, the study is not of non-inferiority or equivalence, as it has not been set up to show this aspect. In fact, the calculation of the Sample Size is determined according to the OSDI, the studies of non-inferiority or equivalence are questionable and should be avoided, when possible, as indicated by many authors (i.e. Garattini S. and Bertelé V. write: “the scientific community should ban non-inferiority and equivalence trials because they are unethical”; also Howick J. (2009), Powers J.H.(2008) etc.) since the margin of error is too subjective and very often there is little attention to ethics and the patient to the advantage of commercial elements. In our case we avoided all this, so if we had opted to use the IOP to calculate the Sample Size we would have had to do a non-inferiority or equivalence study and since the aim of the study was not to show this aspect the analysis of the IOP was only as a corollary to the final results.
- Garattini S. and Bertelé V. Non-inferiority trials are unethical because they disregard patients’ interest. Lancet (2007); 370:1875-7
- Howick J. Questioning the methodologic superiority of ‘placebo’ over ‘active’ controlled trials. The American Journal of Bioethics (2009); 9(9): 34-48
- Powers J.H. Non-inferiority and equivalence trials: Deciphering ’similarity’ of medical interventions. Statistics in Medicine (2008); 27: 343-352
Reviewer 2 Report
Authors have investigated the to evaluate whether the therapeutic switch from a 16 formulation of Bimatoprost 0.1 mg/ml with benzalkonium chloride (BAK) or Bimatoprost 0.3 mg/ml 17 preservative-free to a formulation of Bimatoprost 0.1 mg/ml preservative-free could improve eye surface conditions in patients with glaucoma. For the purpose they measured and analyzed the IOP in both cases. Authors concluded that to evaluate whether the therapeutic switch from a 16 formulation of Bimatoprost 0.1 mg/ml with benzalkonium chloride (BAK) or Bimatoprost 0.3 mg/ml preservative-free to a formulation of Bimatoprost 0.1 mg/ml preservative-free could improve eye surface conditions in patients with glaucoma. This is a clinically intensive study and performed rigorous analysis. Overall, manuscript is presented well. Authors could amend the manuscript based on the following comments to enhance its visibility.
1. Authors may add a comment on (Introduction) methods based on OCT technology to measure the reflectance (directional) of RFNL in vivo as a viable method to detect the glaucoma [1-2]. (https://doi.org/10.1117/12.2190530 ; https://doi.org/10.1117/1.JBO.24.6.066011)
2. Could authors mention the significance of the shared-shapes in Figure 2-3
Author Response
- Authors may add a comment on (Introduction) methods based on OCT technology to measure the reflectance (directional) of RFNL in vivo as a viable method to detect the glaucoma [1-2]. (https://doi.org/10.1117/12.2190530 ; https://doi.org/10.1117/1.JBO.24.6.066011)
Thank you for your suggestion. We have added the required comment.
- Could authors mention the significance of the shared-shapes in Figure 2-3
A violin plot consists of a boxplot and a histogram, but a uniform probability density function(PDF) is used to avoid the subjectivity of binning a histogram, in fact, using PDF provides a more uniform distribution by attenuating the noise. The PDF size/width of violin plot shows the relative frequency with which the values occur. Where the PDF is larger, the value occurs with a higherprobability; while when PDF is narrower, the value occurs less frequently or lower probability.This plot identifies the minimum, first quartile, median, third quartile, and maximum in the same way that a boxplot does. In presence of multimodal data violin plot are a great way of visualizing. Violin plot allow you to see where the data is centered and how it is spread. This plot shows the shape of the data set and is useful for visualizing multiple distributions at once for comparison.

Round 2
Reviewer 1 Report
The authors have responded properly to the comments.